# Mirror-like Bright Al-Mn Coatings Electrodeposition from 1-Ethyl-3 Methylimidazolium Chloride-AlCl_3_-MnCl_2_ Ionic Liquids with Pyridine Derivatives

**DOI:** 10.3390/ma14206226

**Published:** 2021-10-19

**Authors:** Dong Peng, Dalong Cong, Kaiqiang Song, Xingxing Ding, Xuan Wang, Yixin Bai, Xinrui Yang, Changqing Yin, Yuxin Zhang, Jinsong Rao, Min Zhang, Zhongsheng Li

**Affiliations:** 1Southwest Institute of Technology and Engineering, Chongqing 400039, China; pd2019@126.com (D.P.); congdl09@163.com (D.C.); scut_song@163.com (K.S.); sc126789@gmail.com (X.D.); xuan123_wang@sina.com (X.W.); bitbyx@163.com (Y.B.); yxr67938543@163.com (X.Y.); 2College of Material Science and Engineering, Chongqing University, Chongqing 400044, China; CorsacY@163.com (C.Y.); zhangyuxin@cqu.edu.cn (Y.Z.); rjs@cqu.edu.cn (J.R.)

**Keywords:** ionic liquids, electrodeposition, Al-Mn coatings, bright, pyridine derivative additives

## Abstract

The effects of three pyridine derivative additives, 4-hydroxypyridine, 4-picolinic acid, and 4-cyanopyridine, on Al-Mn coatings were investigated in 1-ethyl-3-methylimidazolium chloride-AlCl_3_-MnCl_2_ (EMIC-AlCl_3_-MnCl_2_) ionic liquids. The smooth mirror-like bright Al-Mn coatings were obtained only in the EMIC-AlCl_3_-MnCl_2_ ionic liquids containing 4-cyanopyridine, while the matte Al-Mn coatings were electrodeposited from EMIC-AlCl_3_-MnCl_2_ without additives or containing either 4-hydroxypyridine or 4-picolinic acid. The scanning electron microscope and X-ray diffraction showed that the bright Al-Mn coatings consisted of nanocrystals and had a strong (200) preferential orientation, while the particle size of matte Al-Mn coatings were within the micron range. The brightening mechanism of 4-cyanopyridine is due to it being adsorbed onto the cathode to produce the combined effect of (1) generating an overpotential to promote Al-Mn nucleation; (2) inhibiting the growth of the deposited nuclei and enabling them grow preferentially, making the coating composed of nanocrystals and with a smooth surface. The brightening effect of 4-cyanopyridine on the Al-Mn coatings was far better than that of the 4-hydroxypyridine and the 4-picolinic acid. In addition, the bright Al-Mn coating was prepared in a bath with 6 mmol·L^−1^ 4-cyanopyridine and displayed superior corrosion resistance relative to the matte coatings, which could be attributed to its unique nanocrystalline structure that increased the number of grain boundaries and accelerated the formation of the protective layer of the corrosion products.

## 1. Introduction

Aluminum (Al) and Al alloys exhibit excellent physical properties, especially aluminum–manganese alloys (Al-Mn). Not only do they have low density and good strength, they possess superior hardness and outstanding corrosion resistance, which has attracted much attention in the field of corrosion protection [1,2,3,4,5,6]. Electrodeposition is considered to be the primary technique for the preparation of metal and alloy coatings. Electrodeposition has certain advantages, such as convenient operation, low cost, and low temperature, and it is relatively facile in adjusting the structure and grain size of the deposits compared with spray coatings and physical and chemical vapor depositions [7,8]. However, Al has a high negative reduction potential (−1.67 V vs. SHE), and Al and its alloys can only be electrodeposited in non-aqueous electrolytes [9]. More recently, non-flammable, less volatile nature, wide potential windows, high solubility of the metal salts and high conductivity of room temperature ionic liquids demonstrate their superiority for the deposition of water-sensitive metals, as well as promising alternative electrolytes for the deposition of Al-Mn coatings [10]. In the last decade, ionic liquids have been the focus of fundamental research for the electrodeposition of Al-Mn coatings [3,8,11,12].

The microstructure of the metal coatings has a remarkable impact on its properties, such as brightness, hardness, and corrosion resistance. Lately, a few papers have been published to modulate the microstructure of the Al-Mn electrodeposited from ionic liquids by changing the Mn content, adding appropriate alloying elements, and optimizing the current density. The Schuh group [5] elucidated the effects of Mn content on the Al-Mn coatings’ grain size and phase. It was found that the Al-Mn coatings with a Mn content of 0–7.5 at.% were microcrystalline (7–15 μm) face-centered cubic (fcc) solid solutions and exhibited rough angular surface morphologies. With the Mn content rising between 8.2 and 12.3 at.%, these deposits have a smooth, nodular surface structure and encompass nanometer scale crystals (10–25 nm) of the fcc solid-solution phase coexisting with an amorphous phase. Meanwhile, the hardness of the coatings also increased from 2.8 to 5.2 GPa, compared to the low Mn content (<7.5 at.%). Previous work from the Wang group [13] showed that the Zr added to the Lewis acidic aluminum chloride-1-ethyl-3-methylimidazolium chloride ionic liquids containing 0.04 mmol·L^−1^ MnCl_2_ could obtain dense nanocrystalline Al-Mn-Zr alloy films with superior anticorrosion performance. This could be attributed to the higher overpotential of the electrodeposition induced by ZrCl_4_ in the bath solution. Ling et al. [11] found that the amorphous Al-Mn coating fabricated from MnCl_2_-AlCl_3_-EMIC (0.2 mol·L^−1^ MnCl_2_ in 2:1 AlCl_3_-EMIC) room temperature ionic liquids at a low current density (6 mA·cm^−2^) displayed anodic sacrificial protection and a low corrosion rate for NdFeB. The use of additives was one of the key technologies needed to regulate the microstructure and properties of the electrodeposited coatings. Furthermore, the effects of a large number of additives on the electrodeposition of Al coatings from ionic liquids have been studied, such as nicotinamide [14], tetraethylenepentamine [15], toluene [16], and methyl nicotinate [17]. However, a few investigations report the effects of organic additives on the microstructure and performances of Al-Mn coatings electrodeposited from the ionic liquids.

In the present study, we selected three pyridine derivatives with different functional groups, namely 4-hydroxypyridine, 4-picolinic acid, and 4-cyanopyridine, as organic additives and investigated their effects on the microstructure and properties (e.g., brightness, and corrosion resistance) of Al-Mn coatings electrodeposited from EMIC-AlCl_3_-MnCl_2_ ionic liquid in detail. The brightening mechanism of 4-cyanopyridine was explored.

## 2. Experimental

### 2.1. Preparation of Ionic Liquid Electrolyte

The ionic liquid electrolytes preparation and electrochemical experiments were carried out in an Ar-filled glove box (SG1200/1000TS, Vigor Co., Chengdu, China), in which the moisture and oxygen content was maintained below 1 ppm. The 1-ethyl-3-methylim-idazoliumchloride (EMIC) was synthesized according to Ref. [18]. Anhydrous AlCl_3_ (Alfa Aesar, Tianjin, China, 99.99%) and MnCl_2_ (Rhawn, Shanghai, China, 99.99%) were used as received. The EMIC-AlCl_3_ ionic liquid electrolyte was prepared by mixing EMIC and anhydrous AlCl_3_ at a mol ratio of 1:2. Anhydrous MnCl_2_ (0.04 mol·L^−1^) was added to the pretreated EMIC-AlCl_3_ electrolyte bath before the final addition of each additive and agitated at 333 K for 24 h to achieve the EMIC-AlCl_3_-MnCl_2_ ionic liquid. The following different concentrations (2 mmol·L^−1^, 4 mmol·L^−1^, 6 mmol·L^−1^, and 8 mmol·L^−1^) of the pyridine derivatives were used as additives without further purification: 4-hydroxypyridine (Rhawn, Shanghai, China, 99.86%), 4-picolinic acid (Rhawn, Shanghai, China, 99.00%), and 4-cyanopyridine (Rhawn, Shanghai, China, 98.00%). After the additives were added to the EMIC-AlCl_3_-MnCl_2_ ionic liquid, the electrolyte (EMIC-AlCl_3_-MnCl_2-_additive) was continuously stirred via a magnetic bar (RET basic, IKA, Staufen, Germany) at 333 K until a transparent liquid was obtained.

### 2.2. Electrodeposition of Al-Mn Coatings

Electrodeposition was conducted in a two-electrode cell in the Ar-filled glove box, using a power source (YS9000DDB-1220, Shanghai Yi Sheng Electronics Co., Shanghai, China). Highpurity Al plates (99.999%, 2.5 cm × 3 cm) were used as the anode, and the copper (Cu) plates (1 cm × 4 cm) were used as the cathode. The distance between the anode and the cathode was about 20 cm. Prior to each electrodeposition, the Cu plates were mechanically polished with a 4000# grit waterproof abrasive paper, then ultrasonically cleaned by deionized water and anhydrous ethanol, respectively. After the cleaning, a part of each substrate was covered with PTFE tape so that a square area (1 cm × 1 cm) could be exposed. Al plates were polished and rinsed in sodium hydroxide solution, deionized water and ethanol prior to use, respectively. In order to prevent reoxidation, the dried plates were transferred to the Ar-filled glove box with the least delay possible. The Al-Mn coatings on Cu substrates were electrodeposited with a constant-current mode of 10 mA·cm^−2^ for 30 min. During the electrodeposition, the bath was stirred at 230 rpm and the temperature of the bath was kept at 303 K with a thermostat (RET basic, IKA, Staufen, Germany). After electrodeposition, all coating specimens were immediately transferred from the glove box, washed in succession by deionized water and ethanol, and finally dried with air. The Al-Mn coatings electrodeposited from EMIC-AlCl_3_-MnCl_2_ bath containing no additives, 6 mmol·L^−1^ 4-hydroxypyridine, 6 mmol·L^−1^ 4-picolinic acid, and 6 mmol·L^−1^ 4-cyanopyridine additives were labeled as Al-Mn-BK, Al-Mn-HP, Al-Mn-PA, and Al-Mn-CP, respectively. The schematic diagrams of whole fabrication procedures about the Al-Mn coatings were depicted in Figure 1.

### 2.3. Characterization

A scanning electron microscope (SEM, JSM-7800F, JEOL, Tokyo, Japan) with an energy dispersive spectroscopy (EDS) was used to observe the surface morphology of the Al-Mn coatings. The compositions of various coatings were determined by X-ray diffractometer (XRD, SmartLab-9, RIGAKU Co., Tokyo, Japan) with Cu Kαradiation (λ = 0.15405 nm). The cathodic polarization curves of EMIC-AlCl_3_-MnCl_2_ with and without pyridine derivative additives were investigated using a Princeton Parstat 2273 electrochemical workstation (ARAMTEK Co., Philadelphia, PA, USA) in the glove box at 303 K. The Cu plate (1 cm^2^) was used as the working electrode, and the Al plates (99.999%) were regarded as the reference and counter electrodes. The Raman spectrum was obtained using a LabRam HR Evolution (Jobin Yvon-Horiba, Paris, France) with a 532 nm Ar-Kr 2018 RM laser (Spectra Physics, Milpitas, CA, USA) as the excitation source. The potentiodynamic polarization curves of various Al-Mn coatings were tested using a Princeton Parstat 2273 electrochemical workstation (ARAMTEK Co., Philadelphia, PA, USA), with the coated sample as the working electrode, the platinum plate as the counter electrode, and the saturated calomel electrode (SCE) as the reference electrode. The potentiodynamic polarization curves for all samples were obtained by automatically changing the electrode potential from −0.4 V to 0.6 V with reference to the open circuit potential (OCP) at a sweep rate of 2 mV·s^−1^.

## 3. Results and Discussion

### 3.1. Characterization of the Al-Mn Coatings

The surface morphology of all coated specimens was examined by SEM (Figure 2). The Al-Mn-BK coating consisted of randomly dispersed inhomogeneous angular grains with a diameter of 0.2–1.5 μm (Figure 2a,b). With the addition of 6 mmol·L^−1^ 4-hydroxypyridine to the EMIC-AlCl_3_-MnCl_2_ ionic liquid, a dense Al-Mn-HP deposit with a uniform grain size of approximately 0.5–1.5 μm was observed (Figure 2c,d). When 4-picolinic acid (6 mmol·L^−1^) was used as the additive, the obtained Al-Mn-PA coating exhibited a smoother appearance than the Al-Mn-HP coating, although the grain size of both samples shared a resemblance (Figure 2e,f). Using 6 mmol·L^−1^ 4-cyanopyridine as the additive produced a smoother and flatter Al-Mn-CP deposit with nanocrystals (Figure 2g,h and Appendix A).

The elemental compositions of all Al-Mn coatings are shown in Figure 3, and the corresponding elemental content of the different coatings is listed in Table 1. In the EDS spectrum, all samples exhibited strong Al and Mn peaks. This demonstrates that the Al-Mn coatings were synthesized from the EMIC-AlCl_3_-MnCl_2_ ionic liquid containing no and various pyridine derivative additives. Furthermore, weak O and Cl peaks were found in all the Al-Mn coatings. The small proportion of O originates from the oxidation of the surface of the Al-Mn coatings in atmospheric conditions. The emergence of Cl could be the result of trace amounts of AlCl_4_^−^ or some other chloroaluminate species intercalated into the Al-Mn coatings during the electrodeposition process [19].

Figure 4 reveals the XRD patterns of the Al-Mn-BK, Al-Mn-HP, Al-Mn-PA, and Al-Mn-CP coatings. For the Al-Mn-BK coating, the diffraction peaks of the Al and Cu substrates were confirmed from the XRD patterns. Among them, the four characteristic diffraction peaks (denoted by



) were attributed to the (111), (200), (220), and (311) crystal planes of a typical well-crystallized face-centered cubic (fcc) structure Al materials (JCPDS no. 01-1180) [13]. No evidence of Al-Mn intermetallic formation or metallic Mn was detected in the XRD patterns; however, the EDS data confirmed the presence of Mn in the Al-Mn-BK coated samples and the Al-Mn deposits can be solid solutions of fcc Al. This is in agreement with previous studies [5]. A similar XRD pattern was observed for the Al-Mn-HP coating, and only the intensity of the characteristic diffraction peaks was altered slightly compared to the Al-Mn-BK coating. However, the (220) and (311) characteristic diffraction peaks of the Al-Mn-PA and Al-Mn-CP coatings vanished, and the intensity of the peak on the (200) crystal plane was significantly enhanced. This phenomenon implied that the pyridine derivative additives significantly modified the crystallographic structure of the Al-Mn coatings. The results of the texture calculations for the various Al-Mn coatings obtained from the XRD patterns in Figure 4 are demonstrated in Appendix A. The Al-Mn-BK deposit was preferentially textured, taking the orientation of both the (200) and (220) crystal planes. Nevertheless, the ionic liquid, including various additives, resulted in the Al-Mn-HP, Al-Mn-PA, and Al-Mn-CP coatings having a strongly preferred (200) reflection [20]. A careful comparison of the (200) peaks of the various Al-Mn coatings revealed that the (200) peak of the Al-Mn-CP coating was observably broadened, suggesting that the smaller grain sizes were obtained during the Al-Mn-CP coating (Figure 4). According to the Scherrer equation, the average crystalline size of the Al-Mn-CP deposit was estimated to be 36 nm [21]. Moreover, the determined crystallite size confirmed by XRD was consistent with the SEM micrographs in Figure 2g,h.

The brightness of the Al-Mn coatings was evaluated by observing whether they could reflect the front stripes. The brightness of the metal coatings was related to the smoothness of the surface. As shown in Figure 5a, the Al-Mn-BK coatings obtained from the pure EMIC-AlCl_3_-MnCl_2_ bath presented a dull surface and did not legibly reflect the stripes in front of it. The matte Al-Mn-BK coating composed of Al-Mn particles with a size range of 0.2–1.5 μm resulted in the irregularity of the coating surface with an average roughness of about 243 nm (Appendix A) [15]. The Al-Mn-HP coating remained a lusterless surface (Figure 5b); its average roughness was about 239 nm (Appendix A), which was mainly due to its micron-sized grains. The Al-Mn-PA coating with micron grains reflected the stripes clearer than the Al-Mn-BK and Al-Mn-HP coatings (Figure 5c). This could be attributed to its strong grain preferential orientation, which made the coating surface flatter with an average roughness of approximately 76 nm (Appendix A). The Al-Mn-CP coating exhibited a silvery and mirror-like bright appearance and reflected the stripes (Figure 5d). The SEM images (Figure 2g,h) and the XRD spectrums (Figure 4) of the Al-Mn-CP coating revealed that the 4-cyanopyridine additive led to the coating grain size reaching the nanometer level and possessing a strong (200) crystal plane preferential orientation, which made the coating attained a smooth surface with an average roughness of only 18 nm (Appendix A) [22,23]. Therefore, the Al-Mn-CP coating was bright.

Figure 6 shows the effect of the different concentrations (2 mmol·L^−1^, 4 mmol·L^−1^, 6 mmol·L^−1^, and 8 mmol·L^−1^) of 4-hydroxypyridine, 4-picolinic acid, and 4-cyanopyridine additives on the quality of the Al-Mn coatings electrodeposited at 303 K in EMIC-AlCl_3-_MnCl_2_ bath. The 4-hydroxypyridine and 4-picolinic acid were used as the additive, and only the matte Al-Mn coatings were obtained, revealing that both additives possessed a poor brightening effect on the surface of the Al-Mn coatings. The matte Al-Mn coatings were also electrodeposited in EMIC-AlCl_3_-MnCl_2_-4-cyanopyridine (2 mmol·L^−1^) ionic liquid. However, with an increase in the 4-cyanopyridine concentration to 4 mmol·L^−1^ and greater, the Al-Mn coatings turned from matte to mirror-like bright. Therefore, this result shows that the brightening effect of 4-cyanopyridine was superior to the other two additives. This was due to the stronger grain refinement and crystal plane preferential orientation effects of the 4-cyanopyridine, confirmed with the SEM and XRD results.

### 3.2. The Action of Additives

The cathodic polarization curves of the copper substrates recorded in EMIC-AlCl_3_-MnCl_2_ without and with pyridine derivative additives at 303 K are shown in Figure 7. As shown in Figure 7, the bulk deposition of Al-Mn started at ca.−20 mV vs. Al in the EMIC-AlCl_3_-MnCl_2_ bath solution [14]. When 6 mmol·L^−1^ 4-hydroxypyridine and 4-picolinic acid were added separately as the additives, a slightly negative shift in the onset potential of the deposited Al-Mn was detected. However, the initial deposition potential of the Al-Mn underwent a considerable negative migration in the EMIC-AlCl_3_-MnCl_2_-4-cyanopyridine (6 mmol·L^−1^) ionic liquid, shifting to ca. −120 mV vs. Al. The shift implied that a high overpotential was required to initiate the nucleation and subsequent growth of Al-Mn in the ionic liquid, leading to an increased rate of Al-Mn nucleation, and thus a decrease in grain size. [14]. In addition, it is worth noting that compared with the EMIC-AlCl_3_-MnCl_2_ ionic liquid, the addition of the pyridine derivative additives promoted the decrease in the cathodic deposition current at the identical deposition potential (Figure 7 inset). This indicates that the deposition process of Al-Mn was inhibited by adding these additives. Moreover, the ability of these additives to hinder Al-Mn deposition is as follows: 4-cyanopyridine > 4-picolinic acid > 4-hydroxypyridine.

The Raman spectra and photographs of the EMIC-AlCl_3_-MnCl_2_ ionic liquids with and without pyridine derivative additives are shown in Figure 8. As seen from the Raman spectra of the pure EMIC-AlCl_3_-MnCl_2_ ionic liquid, there are five peaks at 175 cm^−1^, 180 cm^−1^, 310 cm^−1^, 347 cm^−1^, and 432 cm^−1^. Among them, 175 cm^−1^, 310 cm^−1^, and 432 cm^−1^ correspond to the Raman peaks of Al_2_Cl_7_^−^, 180 cm^−1^, and 347 cm^−1^, which belong to the Raman peaks of AlCl_4_^−^ [24]. No new Raman peaks and color shifts of the ionic liquid were detected with the addition of each of the three pyridine derivative additives (6 mmol·L^−1^) to the ionic liquids, indicating that the coordination environment of Al (III) did not vary with the addition of the additives [14,15]. Therefore, the inhibitory effects of 4-hydroxypyridine, 4-picolinic acid, and 4-cyanopyridine additives on the electrodeposition of Al-Mn might be attributed to their adsorption onto the electrode and/or the surface of the Al-Mn deposit [25]. The electron-withdrawing ability of the cyano group in 4-cyanopyridine was stronger than that of the hydroxy group in 4-hydroxypyridineand and the carboxyl group in 4-picolinic acid, which made 4-cyanopyridine more easily adsorbed on the cathode surface. Accordingly, 4-cyanopyridine exhibited the greatest hindering action for the electrodeposition of Al-Mn among the three additives, which is in line with the cathodic polarization curves results in Figure 7 [15].

Based on the double electrical layer’s theory, the schematic diagram of the mechanisms of 4-cyanopyridine in the EMIC-AlCl_3_-MnCl_2_ ionic liquid is displayed in Figure 9. In the EMIC-AlCl_3_-MnCl_2_ ionic liquid, the matte Al-Mn coating was co-deposited smoothly, and there was almost no hindrance during the deposition process. Wang et al. [17] had reported that the active center of pyridine derivative additives is the nitrogen atom of the pyridine ring. On the pyridine ring of the 4-cyanopyridine additive, the cyano group with strong electron-withdrawing ability caused the electron density around the nitrogen atom to decrease, which was conducive to the additive adsorbed onto the cathode when a potential was applied. The adsorbed 4-cyanopyridine primarily had the following two actions: (1) generating overpotential to promote Al-Mn nucleation; (2) inhibiting the growth of the deposited nuclei and making them grow preferentially. Under the combined action, a mirror-like bright Al-Mn-CP coating with nanocrystals and smooth surface was fabricated.

### 3.3. Corrosion Properties of the Al-Mn Coatings

The corrosion resistance of the Cu substrate, Al-Mn-BK, Al-Mn-HP, Al-Mn-PA, and Al-Mn-CP coatings was estimated via potentiodynamic polarization curves in a 3.5 wt. % NaCl aqueous solution, and the corresponding results are displayed in Figure 10 and Table 2. It was found that the Cu substrate exhibited a significant corrosion potential of −239 mV. The Al-Mn coatings electrodeposited onto the Cu substrate surface showed a much more negative corrosion potential than the Cu substrate, revealing that they could have an anodic protective effect on the Cu substrate. Moreover, all Al-Mn coatings displayed the passivation behaviors in the anodic potential region, implying the anodic reaction processes were restrained and could be due to the spontaneous generation of an oxide layer on the surface of the Al-Mn coating [4,26]. Table 2 summarized the Al-Mn specimens’ corrosion potential (*E_corr_*) and corrosion current density (*I_corr_*). The Al-Mn coatings deposited from the EMIC-AlCl_3_-MnCl_2_ ionic liquid had superior corrosion resistance compared to the Cu substrate. In addition, the corrosion resistance of the Al-Mn coatings obtained from the EMIC-AlCl_3_-MnCl_2_-additive was better than that of the Al-Mn-BK coating. In the four Al-Mn coating samples, the Al-Mn-CP coating possessed the lowest corrosion current density (0.98 μA·cm^−2^), which was approximately four and nine times lower than the correspondence corrosion parameter values of the Al-Mn-BK coating (3.81 μA·cm^−2^) and Cu substrate (8.50 μA·cm^−2^), respectively. The excellent corrosion resistance of the Al-Mn-CP coating was due to its nanocrystal structure. Moreover, the nanocrystals increased the number of grain boundaries, which increased the corrosion resistance of the Al-Mn-CP coating [27]. Furthermore, the nanocrystals in the Al-Mn-CP coating induced an increase in the number of active atoms on the surface; this behavior accelerated the formation of the protective layer of the corrosion products. Therefore, the Al-Mn-CP coating had better corrosion resistance than the Al-Mn-HP and the Al-Mn-PA coatings.

## 4. Conclusions

The effects of three pyridine derivative additives with different functional groups, 4-hydroxypyridine, 4-picolinic acid, and 4-cyanopyridine, on the microstructure and properties of the Al-Mn coatings electrodeposited in the EMIC-AlCl_3_-MnCl_2_ ionic liquid were investigated at 303 K. The mirror-like bright Al-Mn-CP coating was successfully fabricated in an EMIC-AlCl_3_-MnCl_2_-4-cyanopyridine ionic liquid. In contrast, matte Al-Mn coatings were obtained in EMIC-AlCl_3_-MnCl_2_ ionic liquids containing 4-hydroxypyridine and 4-picolinic acid. The bright Al-Mn-CP coating consisted of nanocrystals and had a strong (200) preferential orientation, while the matte Al-Mn-HP and Al-Mn-PA coatings had a strong (200) preferential orientation; however, the particle size was in the micron range. The mirror-like bright Al-Mn-CP coating was obtained from the adsorption of 4-cyanopyridine onto the cathode producing the following actions: (1) generating overpotential to promote Al-Mn nucleation; (2) inhibiting the growth of the deposited nuclei and enabling them grow preferentially; making the coating composed of nanocrystals and with a smooth surface. In addition, the Al-Mn-CP coating exhibited a higher corrosion resistance compared to the Al-Mn-BK, Al-Mn-HP, and Al-Mn-PA coatings, which could be attributed to its nanocrystalline structure increased the number of grain boundaries and accelerated the formation of the protective layer of the corrosion products.

## Figures and Tables

**Figure 1 materials-14-06226-f001:**
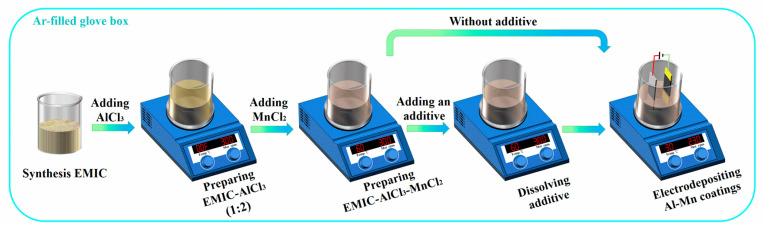
Schematic diagrams about fabrication procedures of the Al-Mn coatings.

**Figure 2 materials-14-06226-f002:**
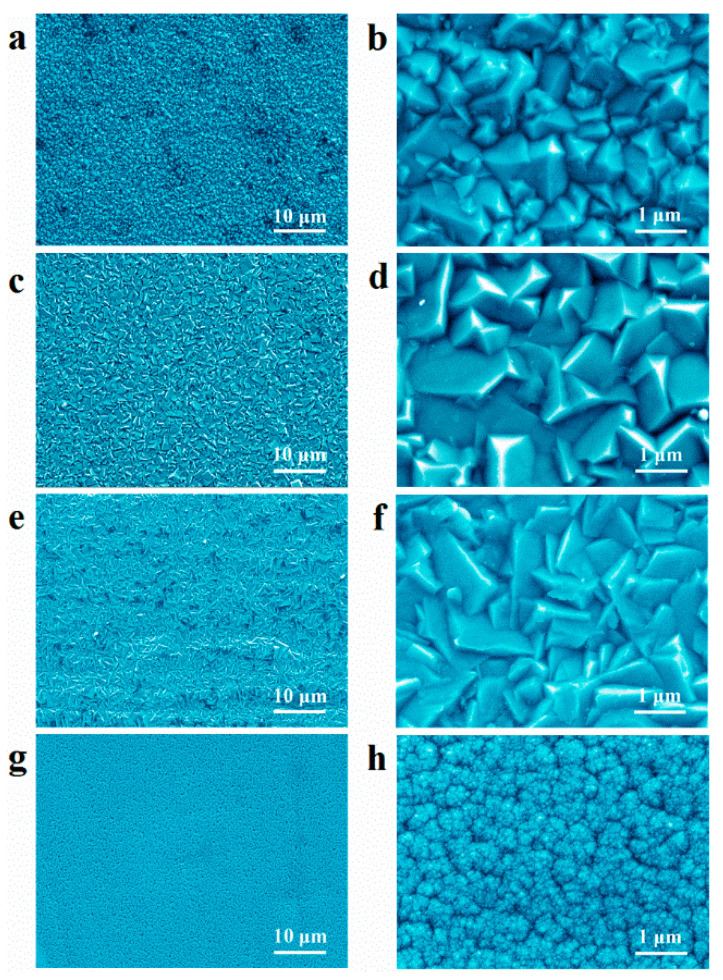
SEM images obtained for the (**a**,**b**) Al-Mn-BK, (**c**,**d**) Al-Mn-HP, (**e**,**f**) Al-Mn-PA, and (**g**,**h**) Al-Mn-CP coatings. (**a**,**c**,**e**,**g**) are the corresponding SEM images of coated samples at a 2000× magnification, (**b**,**d**,**f**,**h**) are the corresponding SEM images of the coated samples at a 20,000× magnification.

**Figure 3 materials-14-06226-f003:**
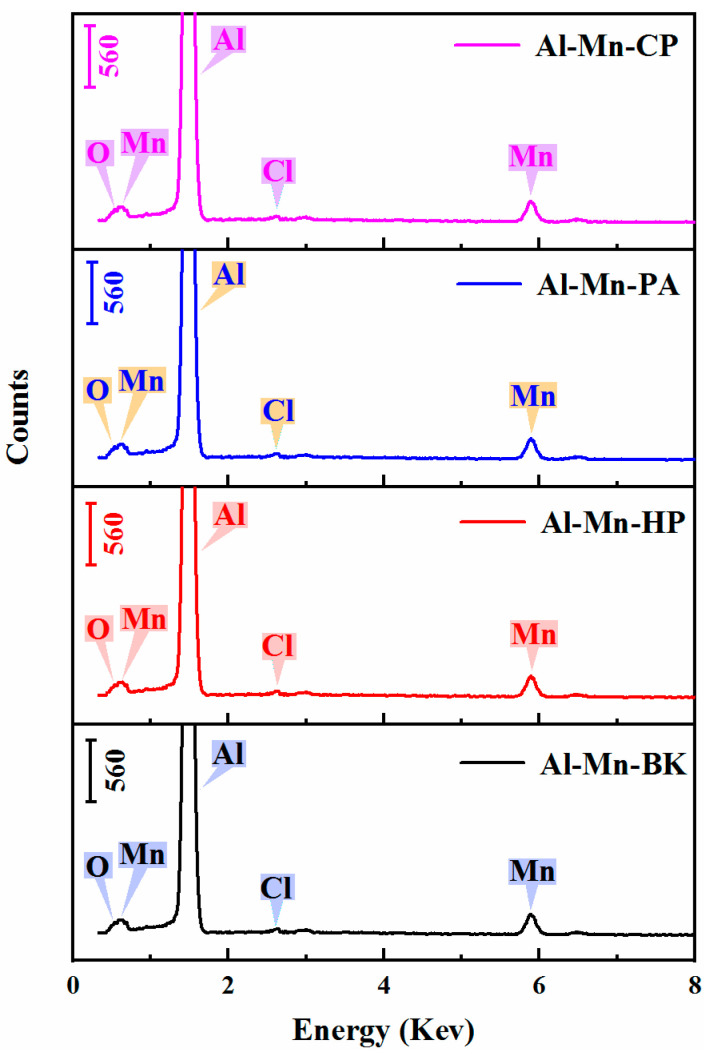
EDS spectrum of various electrodeposited Al-Mn coatings. From bottom to top are the EDS spectra of the Al-Mn-BK, Al-Mn-HP, Al-Mn -PA, and Al-Mn-CP coatings.

**Figure 4 materials-14-06226-f004:**
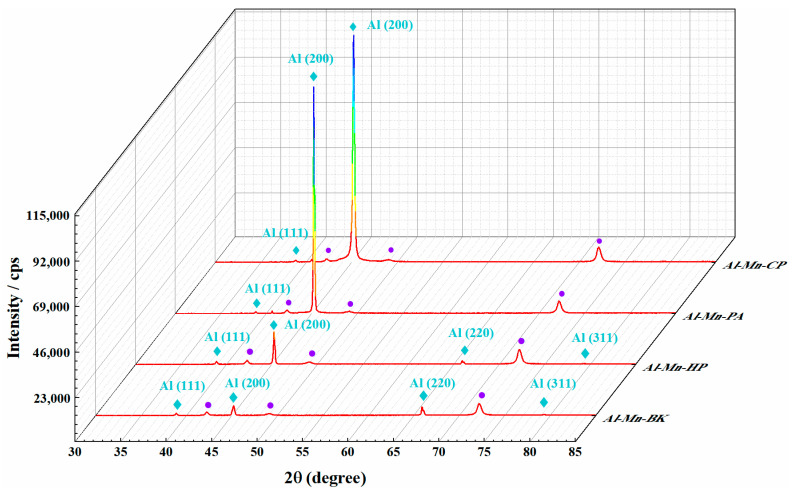
XRD patterns obtained for the Al-Mn-BK, Al-Mn-HP, Al-Mn-PA, and Al-Mn-CP coatings. Diffraction peaks of the Al and Cu substrates were denoted by 

 and 

, respectively.

**Figure 5 materials-14-06226-f005:**
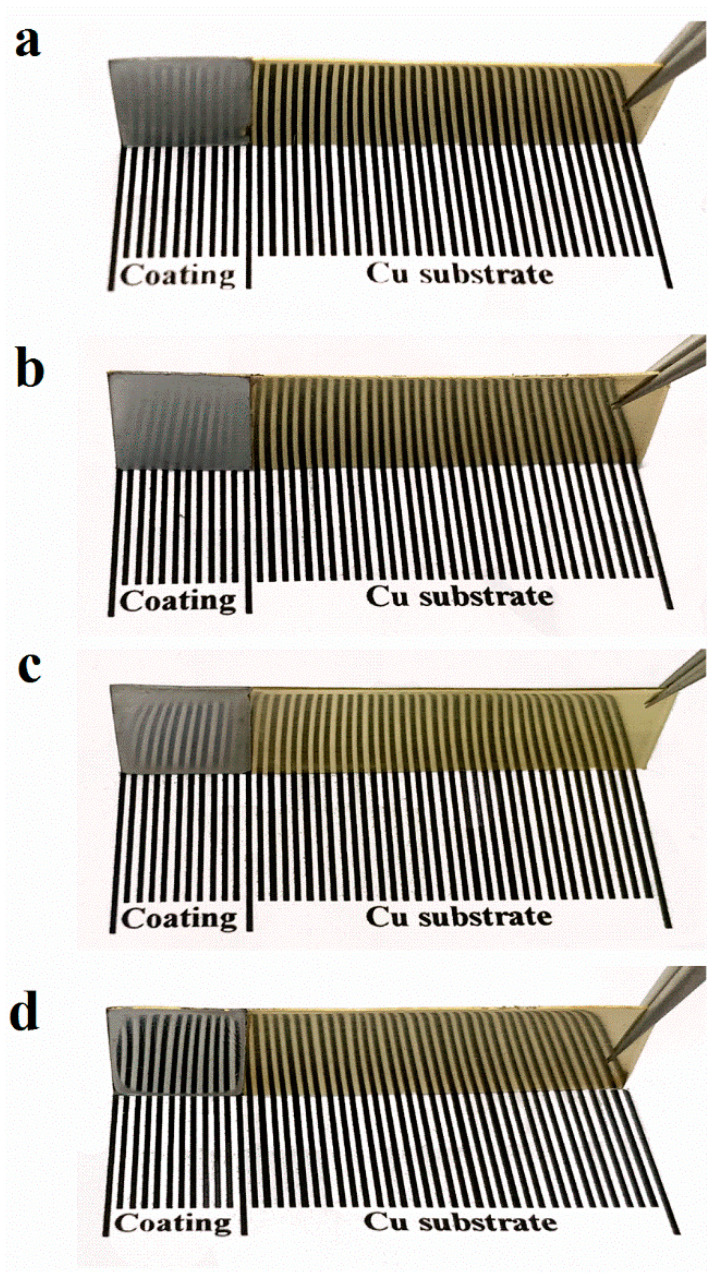
Photographs of the (**a**) Al-Mn-BK, (**b**) Al-Mn-HP, (**c**) Al-Mn-PA and (**d**) Al-Mn-CP coatings reflection stripes.

**Figure 6 materials-14-06226-f006:**
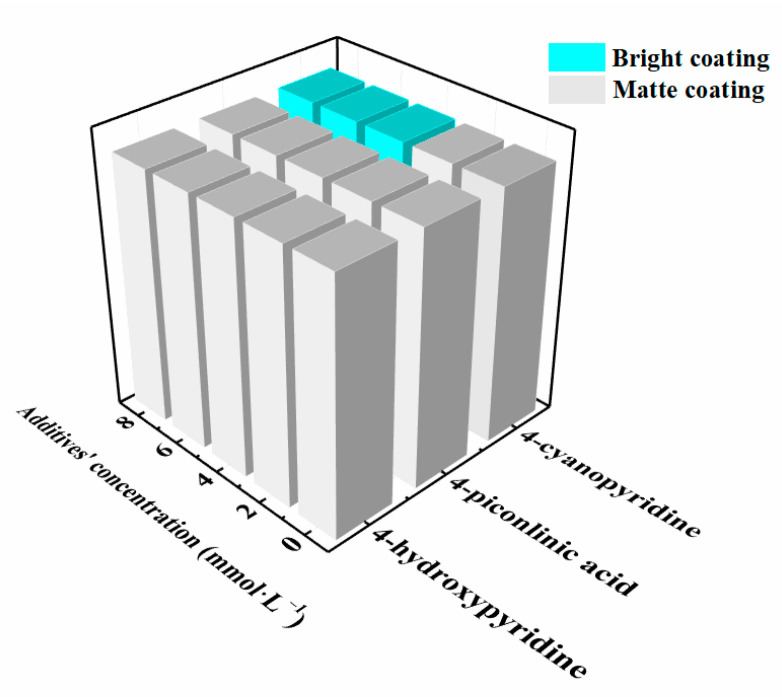
The statistical results of the quality of the Al-Mn coatings electrodeposited at 303 K in EMIC-AlCl_3_-MnCl_2_ containing no, 4-hydroxypyridine, 4-picolinic acid, and 4-cyanopyridine at different additive concentrations (2 mmol·L^−1^, 4 mmol·L^−1^, 6 mmol·L^−1^, and 8 mmol·L^−1^). The cyan blue columns represent that the deposited Al-Mn coatings are bright, and the gray columns represent that the deposited Al-Mn coatings are matte.

**Figure 7 materials-14-06226-f007:**
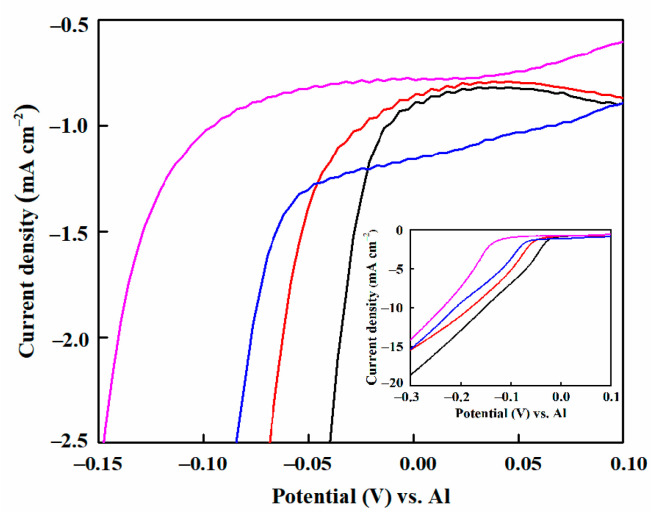
Cathodic polarization curves for electrodeposition of EMIC-AlCl_3_-MnCl_2_ containing no (black line), 6 mmol·L^−1^ 4-hydroxypyridine (red line), 4-picolinic acid (blue line), and 4-cyanopyridine (magenta line) at 303 K on the Cu substrate. Scan rate: 10 mV s^−1^. The inset shows the cathodic polarization curves corresponding to a wider range of horizontal and vertical coordinates.

**Figure 8 materials-14-06226-f008:**
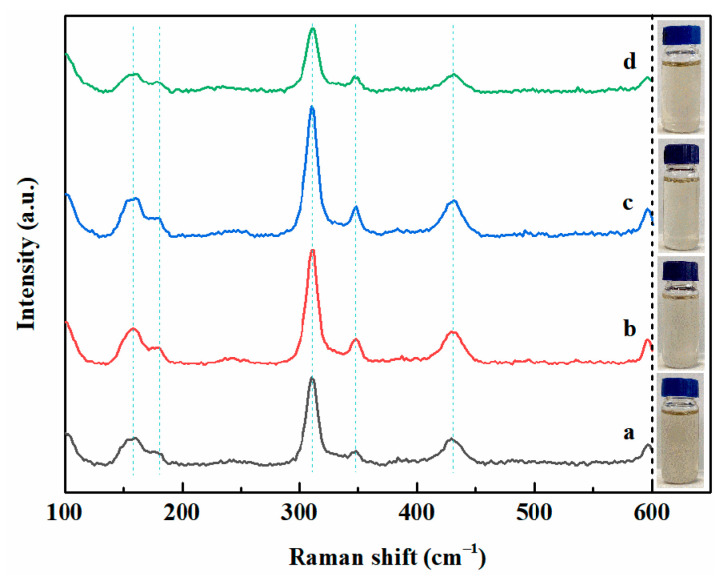
Raman spectra and photographs of the EMIC-AlCl_3_-MnCl_2_ ionic liquids containing (a) no, and 6 mmol·L^−1^ (b) 4-hydroxypyridine, (c) 4-picolinic acid, and (d) 4-cyanopyridine additives.

**Figure 9 materials-14-06226-f009:**
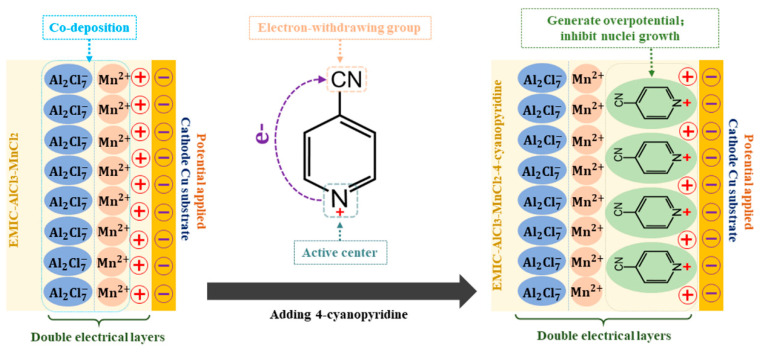
Schematic diagram of the mechanism of 4-cyanopyridine in EMIC-AlCl_3_-MnCl_2_ ionic liquid.

**Figure 10 materials-14-06226-f010:**
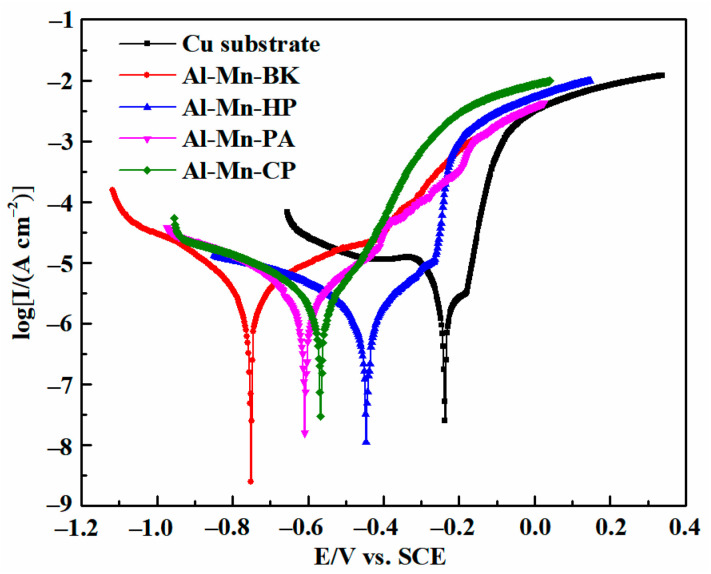
Potentiodynamic polarization curves of the Cu substrate, Al-Mn-BK, Al-Mn-HP, Al-Mn-PA, and Al-Mn-CP coatings in a 3.5 wt % NaCl solution at a scan rate of 1 mV s^−1^.

**Table 1 materials-14-06226-t001:** Elemental compositions of various Al-Mn coatings obtained from the EDS spectra in Figure 3.

Element	Al-Mn-BK	Al-Mn-HP	Al-Mn-PA	Al-Mn-CP
Al (at.%)	90.44	89.38	90.38	90.15
Mn (at.%)	7.49	6.87	7.26	4.85
O (at.%)	1.99	3.31	2.29	2.47
Cl (at.%)	0.08	0.44	0.08	2.53

**Table 2 materials-14-06226-t002:** Corrosion parameters received from the polarization curves in Figure 10.

Materials	*E_corr_* (mV)	*I_corr_* (μA cm^−2^)
Cu substate	−239	8.50
Al-Mn-BK	−747	3.81
Al-Mn-HP	−446	1.72
Al-Mn-PA	−609	3.24
Al-Mn-CP	−578	0.98

## Data Availability

Data sharing is not applicable.

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
