# Peer review of "Mirror-like Bright Al-Mn Coatings Electrodeposition from 1-Ethyl-3 Methylimidazolium Chloride-AlCl3-MnCl2 Ionic Liquids with Pyridine Derivatives"

_materials, 2021, doi:10.3390/ma14206226_

Round 1

Reviewer 1 Report

This paper considers the effect od additives on electrodeposition of Al-Mn coatings with different additives. The tematic of this paper is actual, but presentation of the obtained results is not satisfactory. Authors use very old methods in a discussion of their results; some of them are old about one century, and unsatisfactory for today. For that reason, this form of the paper is not suitable for publishing in this journal. Authors should re-arrange their paper and to include contemporary theories explaining brightness of the coatings. In this moment, I suggest major revision for this paper.

Comments for this paper:

Abstract: What means "directionally"?

The sentence "The combined effect of 4-cyanopyridine ..... ": Comment: please complete with sentence; this form is unclear. The combined effect with what?

The sentence: ".......................... its unique nanocrystalline structure". Please clarify a meaning "its unique nanocrystalline structure". It is better to quantify an improvement of this corrosion resistance with the corresponding data.

The thickness of coatings should be given.

Figure 2: The abbreviations in Figure caption; such as BK, HP, PA and CP, should be given.

The paragraph on page 4 demands rearrangement. You mix in the same paragraph three kinds of results: SEM, XRD and the cathodic polarization curves. After Figure 2, you introduce Figure 4. Please re-arrange it.

Page 5: The sentence: "From Table 1, it is apparent that when the pyridine derivative additives were added to the ionic liquids, the content of Mn in the Al-Mn-HP, Al-Mn-PA, and the Al-Mn-CP coatings were significantly diminished compared to Al-Mn-BK." It is simply not true. With Al-Mn-PA, you have 7.26 at.% Mn, while with Al-Mn-BK, a content of Mn is 7.49 at.%. in Al-Mn-HP, a content of Mn is also very close (6.87 at.%).

Figure 4: For a calculation of Texture coefficients, it is better that Intensity to be given not as a.u., but in cps (count per second, as given by X-ray diffractometer).

Aside from TC coefficients, Authors should also include RTC coefficients (see the original reference: Berube, L.Ph.; Esperance, G.L. J. Electrochem. Soc. 1989, 136, 2314–2315.) in their manuscript. For a clarification to all Readers, Authors should point out that the TC values larger than 1 indicate the existence of the preferential orientation in hkl plane. It is preferably that the Al standard with diffraction peaks to be included on Figure 4.

The sentence: "The grain size of the coatings and the preferential orientation of the crystal plane played a crucial role in the coating’s brightness." This is simply not true. Brightness of the metal coatings can be associated by neither grain size nor the preferential orientation. These are very old theories; some of them are old one century. Authors should give a more precise a definition of coating brightness, in accordance with actual techniques for characterization of metal surface area. Hence, Authors must add fresh references including analysis of brightness of metal coatings.  It is shown for a long time that brightness is not associated with the wavelength of visible light!!!    

Figure 7 and the sentence "The cathodic polarization curves of the copper substrates .... ". Comment: They are not the cathodic polarization curves of the copper substrate but the polarization curves for electrodeposition of EMIC-AlCl3-MnCl2 without and with ....on the Cu substrate.

Authors should represent the polarization curve in the most suitable form. They should give an ordinate up to approximately 2.5 mA cm(-2), not up to 25 mA cm(-2), to see the effect of added additives on the existence and the value of the limiting diffusion current density plateaus. Also, the abscissa should be given between 0.1 and - 0.3 V vs. Al, to see the effect of additives on the cathodic polarization. Authors should discuss results obtained by a re-arrangement of the polarization curves. This is important because for a very long time it is known that additives for the brightness increase the cathodic polarization. Anyway, any effect of additives cannot be seen from the polarization curves shown in the form on Figure 7.

Reviewer 2 Report

Here the authors describe the effects of additives to the electrochemical solution used to make Al-Mn thin films. They provide a thorough analysis, including XRD, SEM, and electrochemical corrosion tests. From this analysis, the authors concluded that the CP additive lead to the best films for this study. While the results are worth publishing, a few questions/suggested edits need to be addressed prior to publication.

Questions/Suggested edits

How many films for each additive were fabricated and analyzed? Are results consistent among the multiple films produced with the same additive (or lack of additive)?

What is the error on the EDS results? How many spots were analyzed and on how many different films prepared in the same way. The difference in Mn for the BK, HP, and PA is rather small. With the CP, could the increased Cl content explain the much lower Mn content in the film?

On page 8, the text regarding Figure 6  indicates that increasing the concentration of 4-piconlinic acid from 2 to 4+ mmol/L resulted in bright films. Do you mean 4-cyanopyridine?

On page 10, do you mean the carboxylic groups on 4-piconlinic acid rather than 4-peaconic acid?

When comparing Table 3 with Figure 10, it seems like the CP voltages do not coincide with each other. From Figure 10, it looks like the CP voltage is around -570 mV, yet on Table 3, the value is -633 mV. 

For the corrosion test results, what potential was chosen for the corrosion current values listed in Table 3? Is it more important to reduce the corrosion current (a kinetic factor) or the potential at which corrosion occurs (thermodynamic factor) or a combination of the two (i.e., resistance calculated based on voltage and current)? One might expect that a lower potential is desired so please comment on which is more desirable and why. 

Round 2

Reviewer 1 Report

Authors adopted mainly all my comments on the original submission, but they must make the two corrections on their Figures:

Figure 4: They should add cps as unit. It is correct that the ordinate to be Intensity / cps

Also, Authors did not adopt my comment for Figure 7. They should reduce an ordinate to be up to approximately 2.5 mA cm(-2).
